# The Persistence of Cross-Reactive Immunity to Influenza B/Yamagata Neuraminidase Despite the Disappearance of the Lineage: Structural and Serological Evidence

**DOI:** 10.3390/ijms26157476

**Published:** 2025-08-02

**Authors:** Yulia Desheva, Polina Kudar, Maria Sergeeva, Pei-Fong Wong, Tamara Shvedova, Ekaterina Bazhenova, Evelyna Krylova, Maria Kurpiaeva, Ekaterina Romanovskaya-Romanko, Vera Krivitskaya, Kira Kudria, Irina Isakova-Sivak, Marina Stukova

**Affiliations:** 1FSBSI ‘Institute of Experimental Medicine’, Saint Petersburg 197022, Russia; polina6226@mail.ru (P.K.); po333222@gmail.com (P.-F.W.); sonya.01.08@mail.ru (E.B.); isakova.sivak@gmail.com (I.I.-S.); 2Medical Institute, St Petersburg State University, Saint Petersburg 199034, Russia; krylova.evelina.03@mail.ru (E.K.); mariakurpyaeva@gmail.com (M.K.); 3Smorodintsev Research Institute of Influenza, Ministry of Health of the Russian Federation, Saint Petersburg 197022, Russia; mari.v.sergeeva@gmail.com (M.S.); romromka@yandex.ru (E.R.-R.); vera.krivitskaya@influenza.spb.ru (V.K.); kira336@yandex.ru (K.K.); marina.stukova@influenza.spb.ru (M.S.); 4Vsevolozhsk Clinical Interdistrict Hospital, Vsevolozhsk 188643, Russia; toma_nn@mail.ru

**Keywords:** influenza B virus, hemagglutinin (HA), neuraminidase (NA), multigene phylogenetic analysis, influenza B virus protein immunology, influenza vaccines

## Abstract

Influenza B viruses, divided into B/Victoria and B/Yamagata lineages, have not had B/Yamagata isolates after 2020. A study evaluated immunity to influenza B surface antigens hemagglutinin (HA) and neuraminidase (NA) in 138 patient sera from 2023 and 23 pairs of sera from 2018 to 2019 vaccine recipients. The phylogenetic tree of the influenza B virus, based on HA and NA genes, shows that the Yamagata lineage evolves gradually, while the Victoria lineage exhibits rapid mutations with short branches. In 2023, mean levels of antibodies to HA and NA of B/Yamagata virus were higher than to B/Victoria, despite no cases of B/Yamagata lineage isolation after 2020. The titers of antibodies to NA of B/Yamagata statistically significantly differed among individuals born before and after 1988. Among patients examined in 2018–2019, neuraminidase-inhibiting (NI) antibody titers before vaccination were higher to B/Yamagata than to B/Victoria, and NI antibodies to B/Victoria and B/Yamagata positively correlated with neutralizing antibodies to B/Victoria virus before and after vaccination. Immunity to B/Yamagata virus was stronger in 2023, despite no isolation since 2020, probably due to the presence of cross-reactive antibodies from B/Victoria infections or vaccinations. Antibodies to NA of B/Victoria and B/Yamagata in 2023 correlated significantly in patients born before 1988, potentially supporting the concept of ‘antigenic sin’ phenomenon for influenza B viruses. The fact that NI antibody titers to B/Victoria and B/Yamagata correlated with neutralizing antibody titers to B/Victoria may suggest broad cross-protection. Studying influenza B virus NA antigenic properties helps understand the evolution and antigenic competition of HA and NA.

## 1. Introduction

Influenza B viruses (IBVs), the family *Orthomyxoviridae*, the genus *Betainfluenzavirus*, cause more than 20% of confirmed influenza cases annually [1], with children and people over 60 years of age being at increased risk of severe disease [2].

Unlike influenza A, which includes 19 HA subtypes and 11 NA subtypes [3,4], influenza B has no subtypes, but is divided into two lines (B/Yamagata and B/Victoria) with only one type of HA and NA. The influenza B virus (IBV) was first identified in the 1940s [5], and antigenic and genetic research in the early 1990s revealed that IBV had split in the 1970s into two distinct antigenic lineages: the B/Victoria/2/1987 and B/Yamagata/16/1988, which have circulated globally since 1985 [6]. These two lineages demonstrate a fluctuating dominance throughout the influenza seasons, likely due to the impact of population immunity [6]. The Yamagata lineage subsequently diverged in 2000 into two antigenic subtypes, clade 2 (B/Massachusetts/02/2012) and clade 3 (B/Wisconsin/1/2010), which circulated in seasonal alternation [7]. The Victoria lineage also underwent a late split: in 2011, it formed two main evolutionary groups (1A and 1B) [7].

An analysis of the circulation of the B/Victoria and B/Yamagata lineages in 2000–2018, that is, before the COVID-19 pandemic, showed that in different seasons in each region of the world, one of the lineages could completely dominate, accounting for from 0% to almost 100% of all cases of influenza B [8]. Although both lineages caused similar numbers of influenza B cases worldwide, their distribution varied depending on various factors. For example, the B/Victoria lineage was more common in tropical countries, while B/Yamagata prevailed in temperate zones of the Northern and Southern Hemispheres. The reasons for this are not entirely clear, but it is possible that different climatic conditions influence the survival and transmission of the viruses. The B/Victoria lineage is able to adapt to the humid and rainy climate of the tropics, while B/Yamagata is probably more resistant to the dry and cold conditions of temperate latitudes [1]. The B/Yamagata lineage was incorporated into the vaccine in the 1998–1999 season. The B/Victoria lineage was included in the influenza vaccines starting in the 2010–2011 season. For a long time, trivalent influenza vaccines were used for active immunoprophylaxis of seasonal influenza. They included the current vaccine strains: two influenza viruses of type A-A(H1N1), A(H3N2), and IBV of one of the two antigenic varieties similar to the reference strains B/Victoria/2/1987 or B/Yamagata/16/1988. However, from 2001 to 2011, in the USA and Europe, in 50% of epidemic seasons, there was a mismatch between the vaccine and epidemic influenza B viruses. Since 2012, quadrivalent influenza vaccines have been used, including two type A influenza viruses, A/H1N1, A/H3N2, and two type B influenza viruses—B/Victoria and B/Yamagata [9].

The results show that B/Yamagata infections predominantly affected older age groups, in contrast to B/Victoria. In the majority of the countries examined, B/Victoria infections were more prevalent among younger individuals, predominantly aged 0–25, with a notable increase among children under the age of 10. Conversely, B/Yamagata infections quite often show a bimodal age distribution, with two main peaks of incidence: one among children under 10 and a second, albeit less pronounced, among adults aged 25–50 [8].

Formation of antibodies to influenza viruses focuses on HA and NA, two critical viral surface proteins. Most antibodies target HA, essential for viral entry into host cells, with those being particularly effective in virus neutralization. However, HA’s high mutation rate limits cross-reactivity across strains [10]. Recent research has highlighted NA-specific antibodies, which inhibit viral egress, thereby reducing replication and disease severity [11]. It is believed that NA undergoes less antigenic variation and shows promise in providing cross-protective immunity [12,13,14]. However, the role of NA antibodies in cross-protection between the B/Victoria and B/Yamagata lineages and the impact of contemporary virus mutations on antibody efficacy remain unclear [15]. Studies of NA immunity to IBV are particularly important, given that the NA inhibitor currently available, Oseltamivir, is the only medication to treat influenza. However, it is less effective in reducing the duration of fever and the presence of the virus in IBV infection compared to IAV infection, particularly in children [16].

It has been shown that antibodies to HA of two antigen lineages of IBV poorly cross-react with each other [9,17]. Earlier, in a study on mice of subunit trivalent inactivated influenza vaccine (IIV), it was found that after the introduction of IIV containing HA of the influenza B virus of the B/Yamagata line, the geometric mean titers (GMT) of hemagglutination-inhibition (HI) antibodies to the vaccine virus were 8 times higher than after the introduction of a similar vaccine containing HA of the Victoria strain. However, when boosting immune mice with IIV, component B from another antigen line (B/Yamagata), it turned out that in the Victoria-Yamagata group, an intense immune response was observed to two antigen lines at once, while in the Yamagata-Victoria group, an immune response was observed only to B/Yamagata viruses. The authors conclude that B/Yamagata lineage viruses are antigenically dominant, while noting a more extensive priming property of the Victoria lineage [13]. In an experimental study of live influenza vaccine, B/Yamagata lineage viruses induced higher levels of post-vaccination antibodies in mice than B/Victoria lineage viruses. Moreover, even in the complete absence of cross-reacting HI antibodies after immunization of mice with B/Victoria virus, protection against infection with B/Yamagata strain was complete, and immunization with B/Yamagata virus against B/Victoria infection was partial [18]. One explanation for such cross-protection is provided by molecular genetic analysis of two antigenic lineages of influenza B viruses. The study suggests that in 2002, a process of reassortment occurred between the two antigenic variants of influenza B viruses. As a result, viruses with the HA gene from the Victorian lineage acquired the NA gene from the Yamagata lineage [13].

The crucial function of the NA in the influenza lifecycle makes it an attractive target for vaccine and treatment development [19]. However, the mechanisms of antigenic competition between HA and NA for influenza B viruses remain poorly understood. Victorian lineage viruses have resurfaced due to significant changes in the HA gene, including nucleotide deletion and coevolution with NA and internal gene segments, while B/Yamagata viruses experience more pronounced seasonal variations caused by antigenic drift of the NA [10]. This highlights the difficulty of predicting which lineage will predominate in the next season, and also explains the high rate of vaccine mismatch, which in tropical countries has been as high as 50% [10].

The fundamental distinctions between the evolutionary processes of influenza A and B viruses in humans have been previously identified. Although the evolutionary rates of the HA, NP, M, and NS genes of influenza B viruses are generally lower than those of their influenza A virus (IAV) components, IBV employs a number of evolutionary mechanisms that contribute to the variability in these viruses. Such mechanisms include genetic divergence into two separate major branches, co-circulation of multiple lineages for long periods of time, the frequent recombination of circulating viruses resulting in new variants with different genome compositions, and systematic deletion and insertion of nucleotides into the HA gene [20]. This was the first instance of a nucleotide deletion and insertion mechanism causing the evolution of viruses into two distinct lineages since 1993. Furthermore, during a span of fifty-seven years, from 1940 to 1997, the inserted and deleted amino acid residues at positions 163, 164, and 165 were consistently Asn-Asp-Asn, respectively. The number of amino acids changes systematically over time. Notably, the HA molecule of the influenza B virus may undergo amino acid substitution in other regions through the above-mentioned mechanisms of deletion and insertion. This could be one of the reasons why the influenza B virus is able to persist for extended periods without undergoing antigenic shift, as seen in IAV [7].

The process of genetic recombination takes place both within and between each influenza B virus strain. The most frequent occurrence is the transfer of genes from the B/Yamagata lineage to the B/Victoria lineage. However, certain regions of the genome remain distinct: the PB1, PB2, and HA segments do not undergo exchange between lineages. At the same time, the majority of contemporary B/Yamagata strains already incorporate NS1/NS2 segments from the Victoria lineage, while the remaining segments (PA, NP, NA/NB, M1/BM2) remain characteristic of the B/Yamagata lineage [21]. Such intricate evolutionary patterns may suggest a symbiotic relationship between distinct viruses, where the timely exchange of gene segments fosters the diversity required for their ongoing circulation [20].

It is possible that the differences in the evolutionary processes of the B/Victoria and B/Yamagata lines are also associated with different mechanisms of interaction of viral HA with cellular receptors. The B/Victoria lineage is able to bind to receptors containing sialic acid in both the α-2,3 and α-2,6 positions, while the B/Yamagata line interacts predominantly with receptors in the α-2,6 position in the human respiratory tract [21].

The aims of this study were to study the phylogeny of influenza B viruses, to compare the molecular structure and activity of NA of modern influenza B viruses, and to evaluate immunity to HA and NA of influenza B viruses among patients of different ages after infection or vaccination.

## 2. Results

### 2.1. Results of Phylogenetic Analysis of Hemagglutinin and Neuraminidase of Influenza B Viruses

The phylodendrogram (Figure 1) compiled from the alignment of two genes (HA and NA) shows that the influenza B virus evolved in two independent directions. The B/Yamagata and B/Victoria lineages have different evolutionary trajectories: the Yamagata lineage has a more gradual branching pattern, with gradual accumulation of changes, while the Victoria lineage has several short branches, which may indicate outbreaks with rapid mutations. RDP4 analysis did not reveal statistically significant recombinations for NA and HA. The PHI test for NA showed a *p*-value of 0.684; for HA, it was a *p*-value of 0.36. Thus, no statistically significant recombinations were found.

### 2.2. Structural Changes and Enzymatic Activity of Neuraminidase of Influenza Viruses B/Victoria and B/Yamagata

Figure 2 shows that there were 22 amino acid differences in the head domain between the NA of influenza B/Brisbane/60/2008 (B/Victoria) and B/Phuket/3073/2013 (B/Yamagata) viruses (Figure 2A,B).

The total number of amino acid differences between the NA of B/Brisbane/60/2008 (B/Victoria) and B/Phuket/3073/2013 (B/Yamagata) was 26, while between the NA of B/Victoria/02/1987 and B/Yamagata/16/1988, the number of amino acid differences was 15 (Figure 2C). At the same time, 27 amino acid differences were detected between the NA of B/Phuket/3073/2013 and B/Yamagata/16/1988, and between the NA of B/Brisbane/60/2008 and B/Victoria/02/1987, there were 32 amino acid substitutions (Figure 2C). It was shown that the enzymatic activity of B/Brisbane/60/2008 (B/Victoria) NA exceeded the activity of B/Phuket/3073/2013 (B/Yamagata) virus NA (*p* < 0.05) (Figure 2D).

### 2.3. Investigating the Presence of Antibodies Against Influenza Viruses of the B/Victoria and B/Yamagata Lineages in Archived Blood Samples from Patients Who Were Tested in 2023

When studying antibodies to influenza viruses of the B/Victoria and B/Yamagata lineages in archival sera of patients examined in 2023, the GMTs of antibodies to HA and NA of the influenza virus belonging to the B/Yamagata lineage were statistically significantly higher than those to HA and NA of the B/Victoria lineage influenza virus (Figure 3A).

The levels of antibodies to NA of influenza B/Victoria viruses did not differ statistically significantly depending on age (Figure 3B). At the same time, titers of antibodies to NA B/Yamagata differed statistically significantly among individuals born before and after 1988 (Figure 3B,C).

Correlation analysis of antibodies to NA of influenza B/Victoria and B/Yamagata viruses in archival sera of patients of different ages examined in 2023 showed that there is a noticeable positive correlation of antibodies to NA of these two viruses in the same sera of individuals born before 1988 (Figure 4A). In individuals born after 1988, the correlation was insignificant (Figure 4B).

At the same time, no significant correlation was observed between antibody titers to NA of B/Victoria and B/Yamagata viruses and the age of the examined patients (Figure 4C). We analyzed the correlation of antibody titers with age in women and men. It turned out that, in men born before 1988, a moderately positive correlation was observed between age and NI titers to B/Victoria and B/Yamagata viruses (r_s_ = 0.38 and r_s_ = 0.35 respectively), and in men born after 1988, a noticeable correlation was found between age and NI titers to B/Yamagata viruses (r_s_ = 0.54) and a moderately positive correlation between age and antibody titers to B/Victoria (r_s_ = 0.42). No such correlations were found in women (Appendix A).

### 2.4. Study of Antibodies to Influenza B Viruses After Vaccination with Seasonal Influenza Vaccines

The patients were immunized during the 2018–2019 influenza season. As in the 2023 case, higher levels of antibodies to the B/Yamagata virus were observed in 2018, i.e., before the emergence of COVID-19. At the same time, the titers of HI antibodies to the B/Victoria and B/Yamagata viruses did not differ, but the titers of NI antibodies and MN antibodies to the B/Yamagata virus were statistically significantly higher than to the B/Victoria virus (Figure 5A).

We observed that only MN antibodies to the B/Victoria virus increased statistically significantly on day 21 after vaccination with seasonal trivalent influenza vaccines (Figure 5A). NI antibodies determined before and after vaccination statistically significantly correlated with neutralizing antibodies to the B/Victoria virus (Figure 5B).

Seroconversions of NI antibodies to the B/Yamagata virus were observed more often than seroconversions of NI antibodies to the B/Victoria virus (Figure 5C). At the same time, the coincidence of the increases in antibodies to the virus of B/Yamagata antigen lineage, determined in other tests, was observed in one case with an increase in HI antibodies and also, in one case, with an increase in neutralizing antibodies (Figure 5C). But there were more coincidences with antibody conversions to another antigen lineage; thus, out of 10 seroconversions of NI antibodies to the B/Yamagata virus, 5 coincided with increases in neutralizing antibodies to the B/Colorado/06/2017 (B/Victoria) virus, and only 1 seroconversion of NI antibodies to the B/Yamagata virus coincided with the seroconversion of NI antibodies to the B/Victoria virus. This shows that NI antibodies to the B/Yamagata virus formed after vaccination with trivalent seasonal vaccines containing the B/Victoria virus were cross-reactive.

It was shown that the most frequently significant increases in MN antibodies were observed for the B/Colorado/06/2017 (B/Victoria) virus after vaccination with the Ultrix and Sovigripp vaccines (Figure 6). At the same time, a moderately positive correlation was noted in the MN antibody titers fold increases to the B/Colorado/06/2017 (B/Victoria) virus and vaccine preparations (r_s_ = 0.44, *p* = 0.03; Appendix A).

## 3. Discussion

Due to the supposed extinction of the B/Yamagata lineage and the relatively slow rate of antigenic drift of IBV, as well as the lack of an animal reservoir, it is believed that IBV can be eradicated from humans [22]. During the COVID-19 pandemic, the circulation of influenza viruses has significantly decreased. Since 2020, for the first time in more than 30 years, viruses of the B/Yamagata antigenic lineage have vanished from circulation [23,24]. The disappearance of the B/Yamagata lineage can be attributed to a combination of factors, including immune pressure, competitive exclusion, and COVID-19 prevention measures [22]. The reduced reproductive rate of the B/Yamagata may have contributed to its vanishing, while the B/Victoria strain has persisted, possibly due to the enhanced fitness of its HA to a more rapid rate of mutation [25]. Strong population immunity against the B/Yamagata lineage may also contribute to the disappearance of this lineage. There is evidence of protective immunity against B/Yamagata in humans due to imprinting by these viruses, where B/Yamagata was apparently the only circulating lineage, which has not been shown for B/Victoria viruses [26]. Nevertheless, it is important to acknowledge that the virus may still be circulating, albeit at a very low level, and may therefore go unnoticed until it has ceased to be present in the population. Antigenic differences of influenza viruses are closely related to genetic changes in HA, which in influenza B viruses is associated with a systematic pattern of nucleotide deletions and insertions, unlike influenza A viruses [7]. Previous phylogenetic studies have revealed inconsistent evolutionary patterns among genes using different methods and data sets [7,20,21]. In this study, we use Bayesian phylogenetic analysis to examine the full-length sequences of two genes common to influenza B viruses, HA and NA. The phylogenetic analysis (Figure 1) performed through the comparison of two genes (HA and NA) shows different evolutionary paths of B/Yamagata and B/Victoria lineages. Analysis showed that surface antigens of influenza B viruses are not subject to frequent recombination between B/Victoria and B/Yamagata lineages. In our study, we used recombinant influenza viruses based on the influenza A/H2N2 virus containing influenza B NA to evaluate antibodies to NA of IBV B/Brisbane/60/2008 (B/Victoria) and B/Phuket/3073/2013 (B/Yamagata). Analysis showed that 26 amino acid differences were detected between the NAs of these two strains, including 22 in the head domain (Figure 4A,B). The difference between these two strains concerned amino acid positions 320 and 343 within the calcium-binding site [27]. The D320K substitution was previously reported in a crucial functional region of the NA protein, specifically in the calcium-binding site situated between amino acids 318 and 350 [28]. And the enzymatic activity of the two strains also differed (Figure 2B) in such a way that the enzymatic activity of NA of the B/Yamagata virus was significantly higher than that of the B/Victoria virus (Figure 2D). The increased amino acid substitution rate on IBV NA between recent strains and the original viruses B/Victoria/02/1987 and B/Yamagata/16/1988 is consistent with previously obtained data that NA is under greater antigenic pressure, emphasizing its crucial impact on IBV evolution [10].

Our study showed that the immunity to HA and NA of the B/Yamagata virus was more intense, despite the fact that B/Yamagata viruses have not been isolated since 2020. A possible explanation for this is the formation of cross-reactive antibodies to the B/Yamagata virus in high titers during natural infection or vaccination with the B/Victoria virus. Previous studies involving B-cell memory and monoclonal antibodies (mAbs) produced by individuals who have been vaccinated with the quadrivalent seasonal vaccine demonstrate the immunological dominance of the B/Yamagata strain HA [29]. The B-cell memory and monoclonal antibodies (mAbs) produced by individuals who have been vaccinated with the quadrivalent seasonal vaccine demonstrate the immunological dominance of the B/Yamagata strain HA. The current immunity against B/Victoria provides strong cross-protection against the Yamagata strain, while immunity against Yamagata offers limited protection against Victoria [15]. This differential cross-protection is due to the antibodies produced by Victoria that target NA, which exhibit cross-lineage reactivity, unlike those produced by Yamagata infections. These results suggest a phenomenon in contemporary influenza that may explain the recent decline of the Yamagata strain, emphasizing the importance of targeting NA in vaccination strategies to enhance cross-lineage protection against influenza [15].

The fact that antibodies to NA of B/Victoria and B/Yamagata showed a noticeable correlation in 2023 in patients born before 1988, and who likely had their first contact with the influenza B virus before the division into two antigenic branches, may indirectly suggest the existence of the phenomenon of ‘antigenic sin’ for influenza B viruses. Studies have shown that B/Yamagata infections mainly affect older people, unlike B/Victoria [8]. Our study showed that in people under 35, the levels of B/Yamagata virus HA and NA antibodies were statistically significantly higher than in older patients.

The protective potential of cross-reactive antibodies to influenza B viruses remains to be determined. While an HI titer of 1:40 or higher has long been considered a marker of immunity, indicating a 50% reduction in the risk of infection [30], no similar marker has been identified for NI and MN titers. For H1N1 and H3N2, HI titers of 1:40 corresponded to MN titers of approximately 1:200 and 1:140, respectively [31]. MN titers not only correlated with protection but also had higher protective efficacy estimates than HI [32]. Studies indicate that a strong response to NA can complement the effects of neutralizing antibodies, potentially leading to better overall protection against influenza infections [33]. It is noteworthy that among patients screened in our study in the 2018–2019 flu season, both B/Victoria and B/Yamagata NA antibodies correlated with B/Victoria virus neutralizing antibodies.

The potential of NA-specific immunity to provide broad cross-protection against IBV in vivo has been demonstrated in several studies [12,13,14,34]. Notably, several studies have demonstrated the effectiveness of NA-based vaccines in protecting mice from both similar and different challenges. For instance, mice that had been immunized with the influenza B virus NA from B/Yamagata/16/88 showed no signs of illness when they were infected with the homologous strain in a sublethal dose. Moreover, they were completely immune to illness and death when they were exposed to a more recent isolate from the Victoria lineage [14]. It was also shown that only the recombinant NA-based Yamagata lineage vaccine, rather than HA recombinant proteins, protected mice against challenge with a significantly antigenically different B/Beijing/243/1997 influenza virus [12]. When administered to mice prior to exposure to a panel of influenza B viruses, the anti-NA sera were as potent as the anti-HA sera in providing protection against homologous challenge and, in some instances, conferred enhanced protection against infection with heterologous influenza B virus strains [35].

The effectiveness of NA-based vaccines has been proven in a guinea pig transmission model [36], where vaccine-induced mucosal immunity against the IBV NA reduced the transmission of heterologous IBV strains. While further human studies are needed to validate these findings, these results indicate that NA-based vaccines may not only provide protection against IBV disease but also potentially prevent the spread of the virus between individuals [37].

Thus, studying influenza B virus NA antigenic properties helps understand the evolution and antigenic competition of HA and NA.

The limitations of this study include the relatively small number of samples. The limitation of using H2NB strains is that HI antibodies to A/H2N2 may be detected in the serum of individuals born between 1957 and 1968 and interfere with the detection of antibodies to NA of influenza B viruses.

## 4. Materials and Methods

### 4.1. In Silico Analysis

The amino acid sequences of HA and NA were obtained from the GISAID (https://gisaid.org/, accessed on 6 December 2024), Influenza virus database (NCBI) (https://www.ncbi.nlm.nih.gov/genomes/FLU/Database/nph-select.cgi?go=database, accessed on 27 March 2025). Multiple alignment of NA and HA sequences using MUSCLE was performed in the Unipro UGENE 64-bit version v49.1 program (http://ugene.net/ru/, accessed on 27 March 2025) [36]. The multigene data set is combined into Sequence Matrix version 1.7.8 (http://www.ggvaidya.com/taxondna/, accessed on 27 March 2025) in Nexus format. The phylogenetic tree was constructed using the Maximum Likelihood method in IQ-TREE (version 3.0.1) (https://iqtree.github.io/, accessed on 5 May 2025). The optimal evolutionary model was determined automatically using the ModelFinder built into IQ-TREE. To assess the robustness of the tree topology, a Bootstrap Analysis was performed with 1000 pseudo-replicants. The tree was visualized in the FigTree program version 1.4.4 (http://tree.bio.ed.ac.uk/software/figtree/, accessed on 5 May 2025), where bootstrap support values were displayed, as well as a scale bar corresponding to the number of evolutionary substitutions per site.

Multiple alignments of HA and NA nucleotide sequences of the studied strains of the B/Victoria and B/Yamagata lineages were checked for the presence of recombinations using the RDP4 (Recombination Detection Program version 4) program (http://web.cbio.uct.ac.za/~darren/rdp.html, accessed on 5 May 2025), where they were loaded in a FASTA file and processed using the Chimaera, MaxChi, BootScan, Siscan, 3Seq and LARD methods. For greater reliability, the multiple alignment of nucleotide sequences was loaded into the SplitsTree program (https://uni-tuebingen.de/fakultaeten/mathematisch-naturwissenschaftliche-fakultaet/fachbereiche/informatik/lehrstuehle/algorithms-in-bioinformatics/software/splitstree/, accessed on 5 May 2025), where the SplitNetwork method and PHI (pairwise homoplasy index test) analysis were used.

### 4.2. Cultivation of Viruses in Developing Chicken Embryos

Chicken embryos (CE) aged 8–10 days (CE were provided by the Poultry Farm Sinyavinskaya, 187326, Leningrad region, Russia) were infected with influenza viruses, for which 200 μL of virus-containing fluid with 6.0 log 10 of the 50% embryonic infectious activity (EID50) was injected into the allantoic cavity using a syringe and incubated at the optimal temperature (33 °C) for 48 h. To determine the hemagglutinating activity of influenza viruses, a series of two-fold dilutions in phosphate-buffered saline (PBS) in a volume of 50 μL was prepared, after which 50 μL of a 1% suspension of chicken erythrocytes was added to the wells. The panel was incubated for 20 min at room temperature, after which the 50% embryonic infectious activity (EID50) was recorded using the Reed and Muench method [37].

### 4.3. Surveyed Contingents

The study used 138 archived blood sera of residents of St. Petersburg and the Leningrad Region of different ages, left over from routine laboratory tests in 2023. None of the subjects had been vaccinated with influenza vaccines during the current influenza epidemic season. The second cohort consisted of 23 pairs of patient sera obtained during an observational study of patients vaccinated with seasonal influenza vaccines, which are based on the vaccine strains recommended by the WHO for the Northern Hemisphere in the 2018–2019 flu season: A/Michigan/45/2015(H1N1)pdm09-likevirus; A/Singapore/INFIMH-16-0019/2016(H3N2)-like virus; and B/Colorado/06/2017 (B/Victoria/2/87 lineage)–like virus. The study included four seasonal trivalent vaccines: a live influenza vaccine (LAIV), two subunit-adjuvanted inactivated influenza vaccines (IIVs)–Grippol and Sovigripp, and a split IIV (Ultrix) [38]. All sera were stored in aliquots at −20 °C so that they were thawed only once before testing.

The study was approved by the Local Ethics Committee at the Institute of Experimental Medicine (protocol № 3/23 dated 20 September 2023) and performed following the tenets of the Helsinki Declaration. All study participants signed written informed consent. After receiving permission from the Ethics Committee, the clinical samples were transferred to the researchers in an anonymized form.

### 4.4. Processing of Blood Serum

To destroy heat-labile hemagglutination inhibitors, all sera were heated at 56 °C for 30 min. To remove NA-sensitive hemagglutination inhibitors, one volume of whole serum was incubated for 18–20 h at 37 °C in the presence of 3 volumes of neuraminidase extract of *Vibrio cholera* (RDE-receptor destroying enzyme, Denka Seiken Co., lot #579081, Tokyo, Japan) followed by heating the samples in a water bath at 56 °C for 1 h. The RDE processing of the blood serum was not carried out if the samples were intended to be used for setting up an enzyme-linked lectin assay (ELLA).

### 4.5. Setting up the Hemagglutination Inhibition Reaction (HI)

The following viruses were used for the HI assay: B/Austria/1359417/2021 (B/Victoria), B/Phuket/3073/2013 (B/Yamagata), and B/Colorado/06/2017 (B/Victoria). The viruses were provided by the Federal State Budgetary Scientific Institution “IEM” (Saint-Petersburg, Russian Federation). To select sera that do not contain HI antibodies to viruses A/H2N2, all samples were analyzed in the HI test with viruses A/Leningrad/134/17/57 (H2N2) and A/Moscow/21/17/65 (H2N2). To eliminate the thermostable hemagglutination inhibitors, the samples were subjected to treatment with a receptor-destroying enzyme (RDE), which is an extract of *Vibrio cholerae* NA (Denka Seiken Co., lot #579081, Tokyo, Japan). For this purpose, the sera were mixed with RDE at a ratio of 1:3 and incubated overnight at 37 °C. After that, the sera were heated for 30′ at 56 °C and 6 volumes of PBS were added to each sample to achieve a final serum dilution of 1:10. RDE-treated and diluted 1:10, blood sera were titrated in short rows of a 96-well polymer plate for immunological reactions with a “U” bottom (82. 1582, Sarstedt, Nuembrecht, Germany) to obtain a series of two-fold dilutions (in 50 μL of FB): 1:10, 1:20, 1:40, 1:80, etc. For this, 50 μL of serum was transferred from the first row to the subsequent rows. Then, a standard dose of virus (8 AE) in a volume of 50 μL was added to each well.

The 100 μL of 0.75% suspension of human erythrocytes of group I (0) was added within 30 min at room temperature and kept for 40 min under the same conditions for erythrocyte sedimentation. The serum titer was determined as the value reciprocal to the dilution of the last well in which there was no hemagglutination.

### 4.6. The Enzyme-Linked Lectin Assay (ELLA)

To evaluate antibodies in ELLA, two H2NB viruses containing NA of influenza viruses of the B/Victoria and B/Yamagata antigenic lineages, presented in Table 1, were used.

The H2NB viruses were prepared by reverse genetics based on the attenuation donor A/Leningrad/134/17/57 (H2N2) and the epidemic influenza virus B/Brisbane/60/2008 (B/Victoria) or B/Phuket/3073/2013 (B/Yamagata). The H2NB viruses were assembled using an 8-plasmid system according to the previously developed protocol [39] possessing a genome formula of 7:1, where the genes encoding one of the surface proteins-HA, and internal and non-structural proteins (PB2, PB1, PA, NP, M, NS) belong to the donor of attenuation A/Leningrad/134/17/57 (H2N2); the gene encoding NA is a chimeric genetic construct (Appendix A). The H2NB viruses were concentrated and purified in a sucrose step gradient. After that, 96-well plates (650001, Greiner Bio-One, Kremsmünster, Austria) were coated with fetuin at a concentration of 50 μg/mL (Sigma-Aldrich, lot #SLCD7337, St. Louis, MO, USA) overnight at +4 °C. After that, 60 μL of the studied sera, which had been heated at 56 °C for 30 min, were serially diluted with phosphate-buffered saline containing bovine serum albumin (PBS-BSA) and incubated with an equal volume of pre-diluted virus at a working dose of 64–128 hemagglutinating units (HAU) per 0.1 mL for 30 min at 37 °C. After incubation, 100 μL of the mixture was applied to fetuin-coated wells of the plates, after 3-fold washing. After incubation for 1 h at 37 °C, the plates were washed and NA activity was assessed by incubation with peroxidase-labeled peanut lectin (2.5 μg/mL, L7759, Sigma-Aldrich, St. Louis, MO, USA) for 1 h at room temperature, followed by washing and adding of 100 μL peroxidase substrate 3,3′,5,5′-tetramethylbenzidine (TMB). The reaction was stopped after 5 min by adding 100 μL of 1 N sulfuric acid. Optical density (OD) values were measured at 450 nm using a microplate reader (Elx800, Bio-Tek Instruments Inc., Winooski, VT, USA). The titer of serum NA-inhibitory antibodies was assigned as the reciprocal dilution of the sample with 50% inhibition of NA activity.

### 4.7. Microneutralization Test (MN)

Microneutralization assay was performed using MDCK London Line cell culture obtained from International Reagent Resource (cat. no FR-58) within passages 25–40 (main bank frozen at 8/8/4 passage (CDC/IRR/RII)). For the MN setup, a monolayer of MDCK cells was grown in 96-well flat-bottom polystyrene plates for adherent cultures (Sarstedt, Nümbrecht, Germany). After removing the maintenance medium and washing the plates twice with phosphate-buffered saline, 50 μL of falling two-fold dilutions (starting with a 1:10 dilution) of blood serum samples in DMEM containing trypsin TPCK at a concentration of 2 μg/mL were added. Then, 50 μL of a standard dose of the virus B/Colorado/06/2017 (B/Victoria) (200 TCID50/0.05 mL = 100 TCID50/0.1 mL), diluted to the indicated concentration with the same medium, was added to each well. The plates were incubated in a CO_2_-fed thermostat at 34 °C for 72 h. Inhibition of virus reproduction was determined by a hemagglutination assay with 0.75% suspension of chicken erythrocytes.

Seroconversion to influenza antigens was defined as a fourfold increase in HI and MN antibody titers and a twofold increase in NI antibody titers. All samples were analyzed in duplicate.

### 4.8. Statistical Processing of Data

The results were processed using the statistical package “GraphPad” (version 8.4.3) and the statistical package “Statistica” (version 6.0). The geometric mean titers (GMT) were used to express the average values of antibody titers. The normality of distribution was studied using D’Agostino & Pearson omnibus normality test. When comparing samples that did not meet the assumptions of normal distribution of the dependent variable within each group and homogeneity of variance, a nonparametric criteria were used (Mann–Whitney, Wilcoxon signed ranks). In the case of nominal data, a two-sided version of Fisher’s exact test was used. A nonparametric measure of the statistical relationship between two variables was performed using Spearman’s rank correlation coefficient (r). The null hypotheses tested by the criteria were rejected at *p* < 0.05.

## 5. Conclusions

Molecular genetic analysis of influenza B viruses for two genes-HA and NA-showed that the surface antigens of influenza B viruses have different evolutionary paths between the two lineages and are not subject to noticeable recombination between B/Victoria and B/Yamagata. It was shown that in 2023, the geometric mean titers of antibodies to HA and NA of the influenza virus of the B/Yamagata lineage were statistically significantly higher than those to HA and NA of the influenza virus of the B/Victoria lineage. Antibody titers to B/Yamagata NA showed statistically significant differences among individuals born before and after 1988. Among patients examined in 2018–2019, antibodies to NA from B/Yamagata were higher before vaccination than to NA from B/Victoria, and NI antibodies to B/Victoria and B/Yamagata correlated with neutralizing antibodies to B/Victoria virus.

## Figures and Tables

**Figure 1 ijms-26-07476-f001:**
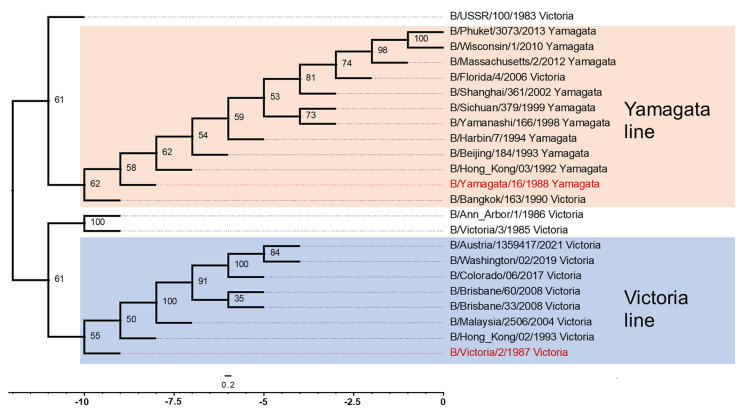
Phylogenetic tree based on the concatenated sequences of the HA and NA genes of influenza B virus (Victoria/Yamagata). The confidence scores for nodes are indicated by ML (Maximum Likelihood) and BS (Bootstrap Analysis). The strains that are the central focus of the study are highlighted in red. The two main lineages into which the group B virus has diverged are indicated by pink and blue shading. The scale bar corresponds to 0.2 expected substitutions per site. HA alignment from 1 to 585 amino acids; NA alignment from 1 to 466 amino acids. The historically important strains B/Yamagata/16/1988 and B/Victoria/2/1987 are highlighted in red and are used as references for their lineage. The reliability of the branching patterns is confirmed by high bootstrap values.

**Figure 2 ijms-26-07476-f002:**
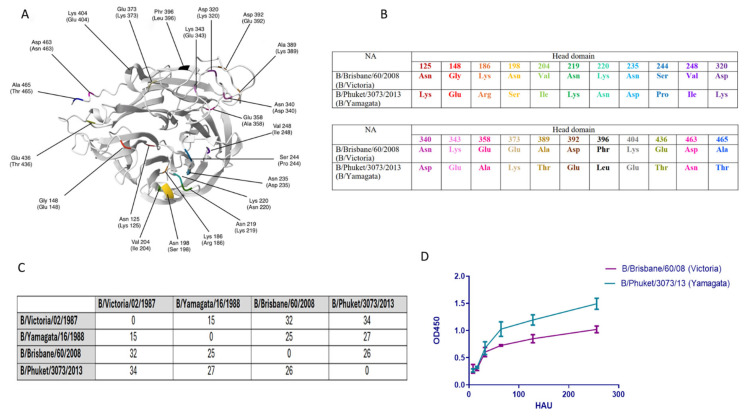
Molecular analysis of NA of B/Brisbane/60/2008 (B/Victoria) and B/Phuket/3073/2013 (B/Yamagata) viruses. (**A**) Spatial model from 77 to 466 amino acids built in the RCSB PDB modeling program (https://www.rcsb.org/, accessed on 6 December 2024), the structure of the model is Cartoon. Differing amino acids are highlighted in colors. (**B**) Amino acid differences in the head domain of NA B/Brisbane/60/2008 (B/Victoria) and B/Phuket/3073/2013 (B/Yamagata) viruses. (**C**) Total number of amino acid substitutions in NA of influenza viruses B/Victoria and B/Yamagata. (**D**) Dependence of the result of the enzymatic reaction on the concentration of test viruses H2NB-Brisbane and H2NB-Phuket, expressed in hemagglutinating units (HAU).

**Figure 3 ijms-26-07476-f003:**
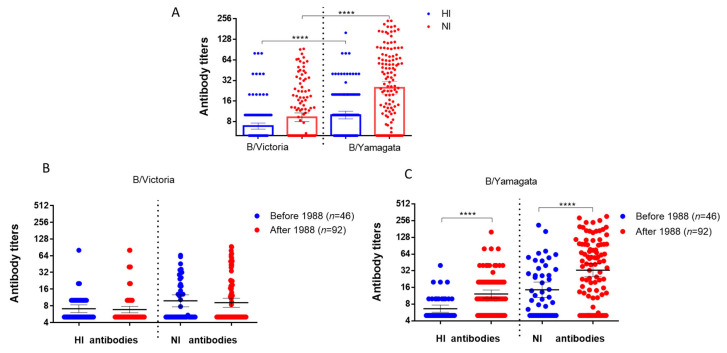
Results of the study of antibodies to surface antigens of influenza viruses of the B/Victoria and B/Yamagata lineages among patients examined in 2023. The geometric mean titers (GMT) and 95% confidence intervals are presented. **** *p* < 0.0001, Mann–Whitney test. (**A**) Levels of antibodies to HA and NA of influenza B viruses among 138 patients were analyzed. (**B**) The GMTs of antibodies to the surface antigens of the B/Victoria lineage among patients of different ages. (**C**) The GMTs of antibodies to the surface antigens of the B/Yamagata lineage among patients of different ages.

**Figure 4 ijms-26-07476-f004:**
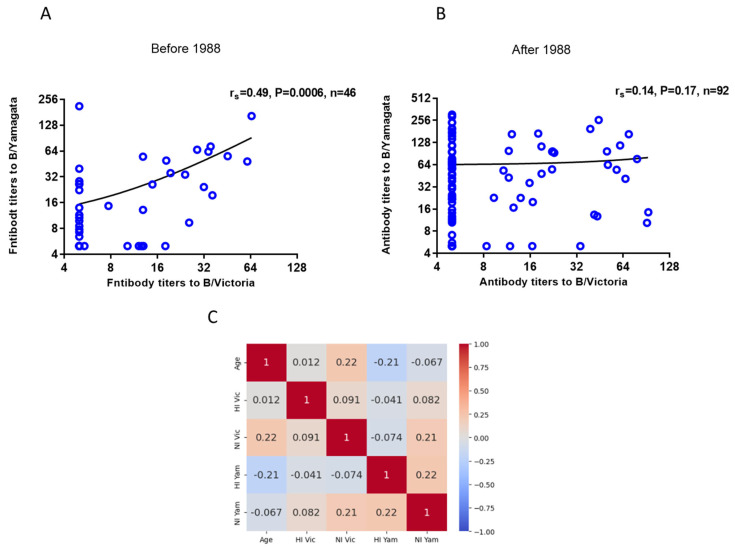
Correlation analysis of antibody titers to NA of influenza B/Victoria and B/Yamagata viruses detected in ELLA among patients of different ages examined in 2023. Spearman correlation coefficients are shown. (**A**) Group of patients born before 1988 (*n* = 46). (**B**) Group of patients born in 1988 and later (*n* = 92). (**C**) Correlation analysis in the general group of patients (*n* = 138). The data were standardized using the Z-score method. The pattern of feature intensity was generated using the built-in functions of the Seaborn library in Python 3. The cells contain the values of the Spearman correlation coefficient (r_s_). The level of the correlation was determined as follows: r_s_ < 0.3—low correlation, r_s_ = 0.3–0.49—moderate correlation, r_s_ = 0.5–0.69—noticeable correlation, r_s_ ≥ 0.7—high correlation.

**Figure 5 ijms-26-07476-f005:**
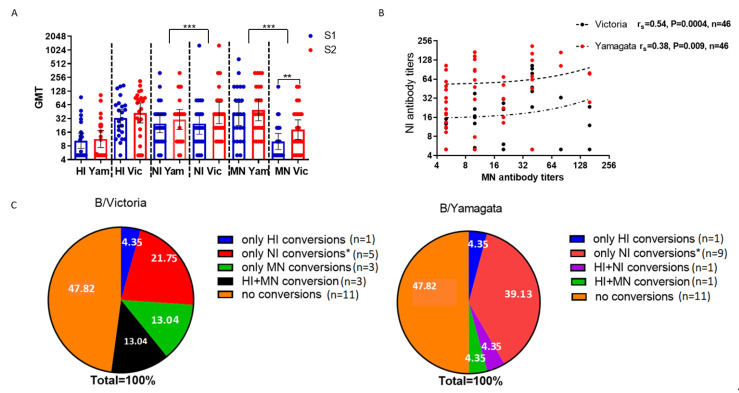
Analysis of antibodies to influenza B/Victoria and B/Yamagata viruses in paired sera of patients vaccinated with seasonal influenza vaccines (*n* = 23). (**A**) Antibody titers; S1—before vaccination, S2—on day 21 after vaccination. Geometric mean antibody titers and 95% confidence intervals are shown. **—*p* < 0.01; ***—*p* < 0.001, Wilcoxon signed ranks test. (**B**) Correlation analysis of titers of neuraminidase-inhibiting and neutralizing antibodies to B/Victoria virus. Spearman correlation coefficients are shown. Analysis of all sera obtained both before and after vaccination. Each point represents an individual serum. (**C**) Proportions of individuals with seroconversion of antibodies to B/Victoria and B/Yamagata viruses. *—the proportion of individuals with seroconversion of only NI antibodies to B/Victoria virus is statistically significantly lower than to B/Yamagata virus (*p* = 0.009), Fisher’s exact test.

**Figure 6 ijms-26-07476-f006:**
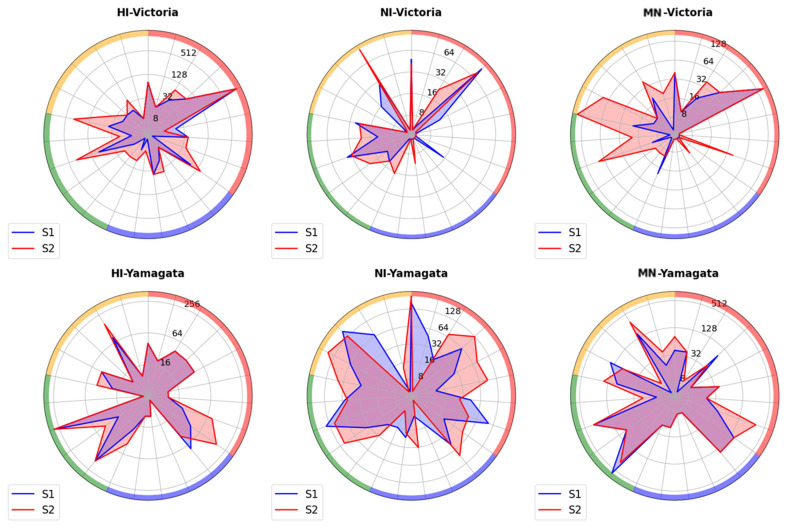
Spider spots diagrams presenting the immune response to B/Victoria and B/Yamagata virus antigens in paired sera of patients vaccinated with seasonal influenza vaccines (*n* = 23), depending on the vaccine preparation. For each vertex antibody, the colors around the circle represent individual influenza vaccines: purple—LAIV, blue—Grippol plus, green—Sovigripp, yellow—Ultrix. Each vertex represents individual titers of HI, NI, and MN antibodies expressed in log2 before vaccination (blue, S1) and on day 21 after vaccination (red, S2).

**Table 1 ijms-26-07476-t001:** Origin of HA and NA genes of recombinant H2NB viruses.

No.	Name	Origin of HA	Origin of NA
1	H2NB-Brisbane	A/Leningrad/134/17/57 (H2N2)	B/Brisbane/60/2008 (B/Victoria)
2	H2NB-Phuket	A/Leningrad/134/17/57 (H2N2)	B/Phuket/3073/2013 (B/Yamagata)

## Data Availability

The original contributions presented in this study are included in the article/Appendix A. Further inquiries can be directed to the corresponding authors.

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
