# Peer review of "The Persistence of Cross-Reactive Immunity to Influenza B/Yamagata Neuraminidase Despite the Disappearance of the Lineage: Structural and Serological Evidence"

_ijms, 2025, doi:10.3390/ijms26157476_

Round 1

Reviewer 1 Report

Comments and Suggestions for Authors

The study investigates the persistence of cross-reactive immunity to influenza B virus neuraminidase (NA) of the B/Yamagata lineage despite its disappearance from circulation since 2020. By analyzing archived patient sera and sera from individuals vaccinated with seasonal influenza vaccines, the research reveals higher antibody levels against B/Yamagata NA compared to B/Victoria NA. The phylogenetic analysis of hemagglutinin (HA) and NA genes shows distinct evolutionary paths between the two lineages. Notably, antibodies to B/Yamagata NA correlate significantly with neutralizing antibodies to B/Victoria, suggesting broad cross-protection. The study highlights the importance of NA in providing cross-lineage immunity and the potential of NA-based vaccines in influenza prevention. However, the manuscript has some issues that need to be addressed.

  1. The abstract is too long and needs to be concise.
  2. The subheading of the results section needs to be rewritten to highlight the findings of the research.
  3. Figure 3B lacks statistical analysis; please indicate if there is no statistical difference.
  4. Please check the manuscript's format and ensure that the references comply with the journal's requirements.

Author Response

We thank the Reviewer for your attentive attitude towards our work.

  1. The abstract is too long and needs to be concise.

Response 1. We thank the Reviewer for this comment. We have shortened the abstract to 250 words.

  1. The subheading of the results section needs to be rewritten to highlight the findings of the research.

Response 2. We thank the Reviewer for this remark. We have corrected the subheadings.

  1. Figure 3B lacks statistical analysis; please indicate if there is no statistical difference.

Response 3. We thank the Reviewer for this comment. We added: “…did not differ statistically significantly depending on age…”.

  1. Please check the manuscript's format and ensure that the references comply with the journal's requirements.

Response 4. We thank the Reviewer for this comment.

Reviewer 2 Report

Comments and Suggestions for Authors

The presented material is interesting for a wide audience, since the theoretical basis of the vaccination strategy in relation to influenza B represents an important area of immunology. The authors identified a difference in the development of influenza B/Victoria and B/Yamagata viruses and their mutations. An important fact presented as a result of the studies is the differences in the level of NA B/Yamagata antibodies in the population older and younger than 37 years. The results revealed during the study of neuraminidase-inhibiting antibody are significant as broad cross-protection is shown. Data on the antigenic properties of influenza B virus NA allows us to judge the evolution and antigenic competition of hemagglutinin and neuraminidases.

Authors provided sufficient references to understand the gap which the presented research fills. Archived blood sera of 138 residents of St. Petersburg and the Leningrad Region of different ages are analyzed. Molecular analysis is performed at a modern level, modeling programs as well as statistical package are applied. Recombinant influenza viruses involved in the study are described well. Correlation analysis is described logically; experimental data on the sera of patients immunized during the 2018-2019 influenza season, analysis of antibodies is provided, the origin of viruses is reported. Seroconversions of neuraminidase-inhibiting antibodies to the B/Yamagata virus were observed more often than seroconversions of neuraminidase-inhibiting antibodies to the B/Victoria virus. It was shown that surface antigens of influenza B viruses are not subject to frequent recombination between B/Victoria and B/Yamagata lineages. Diagrams presenting the immune response are perfect.

All necessary information on in silico study, cultivation of viruses, blood serum processing, microneutralization assay is given, the approach to recombinantH2NB viruses is described. Conclusions are concise.

The manuscript can be published after minor revision. Please fix some points:

Line 18. Extra (

Line 602. 275-299

Line 616. 852-863

Line 657. 1191-1199

Author Response

We thank the Reviewer for the positive assessment of our work and for useful comments.

Comment 1. Line 18. Extra (

Response 1. We thank the Reviewer for this comment. We fixed it

Comment 2. Line 602. 275-299

Response 2. We thank the Reviewer for this comment. Corrected.

Comment 3. Line 616. 852-863

Response 3. We thank the Reviewer for this comment. Corrected.

Comment 4. Line 657. 1191-1199

Response4. We thank the Reviewer for this comment. Corrected.